# Cell Biology of Giant Cell Tumour of Bone: Crosstalk between m/wt Nucleosome H3.3, Telomeres and Osteoclastogenesis

**DOI:** 10.3390/cancers13205119

**Published:** 2021-10-13

**Authors:** Ramses G. Forsyth, Tibor Krenács, Nicholas Athanasou, Pancras C. W. Hogendoorn

**Affiliations:** 1Department of Pathology, University Hospital Brussels (UZB), Laarbeeklaan 101, 1090 Brussels, Belgium; p.c.w.hogendoorn@lumc.nl; 2Labaratorium for Experimental Pathology (EXPA), Vrije Universiteit Brussel (VUB), Laarbeeklaan 103, 1090 Brussels, Belgium; 31st Department of Pathology and Experimental Cancer Research, Semmelweis University, Üllöi ut 26, 1085 Budapest, Hungary; krenacst@gmail.com; 4Department of Histopathology, Nuffield Orthopaedic Centre, University of Oxford, NDORMS, Oxford OX3 7HE, UK; nick.athanasou@ouh.nhs.uk; 5Department of Pathology, Leiden University Medical Center (LUMC), Albinusdreef 2, 2300 RC Leiden, The Netherlands

**Keywords:** giant cell tumour of bone, osteoclast, mononuclear spindle-shaped cell, telomeric association, G34W

## Abstract

**Simple Summary:**

The overall clinical and cell biological behaviour of giant cell tumour of bone still remains very difficult to understand. This entity is a prototype neoplasm illustrating that the scientific work around nucleosome mutations in cancer is a hot topic. This review tries to summarise and integrate the wide spread insights in this research era with specific attention to the cell biological aspects of giant cell tumour of bone with the focus on genomic and epigenomic alterations that may lead to genomic instability and malignancy. Moreover, the ultimate goal is to connect all major characteristics of this tumour to a comprehensive understanding, like parts of a broad etiogenic puzzle, which may lead to new insights.

**Abstract:**

Giant cell tumour of bone (GCTB) is a rare and intriguing primary bone neoplasm. Worrisome clinical features are its local destructive behaviour, its high tendency to recur after surgical therapy and its ability to create so-called benign lung metastases (lung ‘plugs’). GCTB displays a complex and difficult-to-understand cell biological behaviour because of its heterogenous morphology. Recently, a driver mutation in histone H3.3 was found. This mutation is highly conserved in GCTB but can also be detected in glioblastoma. Denosumab was recently introduced as an extra option of medical treatment next to traditional surgical and in rare cases, radiotherapy. Despite these new insights, many ‘old’ questions about the key features of GCTB remain unanswered, such as the presence of telomeric associations (TAs), the reactivation of hTERT, and its slight genomic instability. This review summarises the recent relevant literature of histone H3.3 in relation to the GCTB-specific G34W mutation and pays specific attention to the G34W mutation in relation to the development of TAs, genomic instability, and the characteristic morphology of GCTB. As pieces of an etiogenetic puzzle, this review tries fitting all these molecular features and the unique H3.3 G34W mutation together in GCTB.

## 1. Cellular Makeup of GCTB

Giant cell tumour of bone (GCTB) consists of four different morphological cellular components, each presenting with its own intrinsic and extrinsic programmed signalling pathways (Figure 1 and Figure 2). The different types of interactions between these components and the overall cell biological and clinical behaviour of this tumour is largely not well understood. Consequently, mechanisms of tumour growth, invasion and metastatic potential remain hard to predict [1,2,3]. The first morphological component of this tumour consists entirely of non-osteoclastic macrophages which do not express CD14 or CD33 [4]. The upshot of macrophage-colony-stimulating factor (M-CSF) production by osteoclast-committed macrophages, pre-osteoclasts and large osteoclastic giant cells is that these non-osteoclastic macrophages are attracted to the tumour microenvironment. While pre-osteoclasts and large osteoclastic giant cells express CD33, CD51 and RANK, this is not the case for CD14, CD163 and HLA-DR [5]. Osteoclast-committed macrophages do express CD14 and/or CD33 together with RANK on their membrane in GCTB. The fate of this second component is taking part in osteoclastogenesis, promoted by the RANKL/RANK pathway, therefore contributing to the osteolytic character of GCTB [6,7,8]. Though these giant cells form by fusion of immature monocytic precursors, they show early replication activity aborted by cyclin-dependent kinase inhibitors at G1-phase [9]. The third GCTB component comprises different types of extracellular matrix proteins, their interlaying fibroblastic cells and a dense network of leaky blood vessels. This tumour microenvironment supports tumour growth and expansion by modulation of its physical characteristics, enabling cell mobility and cellular communication. This is clearly illustrated by GCTB patients presenting with metastatic lung noduli. The primary GCTB associated with such metastatic tumours typically shows stromal haemorrhages and formation of thrombi [10]. In agreement with these observations, the reduced expression of lumican and decorin and the elevated levels of the tissue-remodelling- and cancer-invasion-related tenascin C, both reflect impaired extracellular matrix integrity, and correlate with local relapse and metastatic disease [11,12]. The fourth morphological component includes the true neoplastic cells, which are also of osteoblastic origin (mononuclear spindle-shaped cells (MSCs)). These cells display a high mitotic activity of up to 20 mitoses per 10 high-power fields and their accelerated cell cycle progression is linked to increased risk of disease progression. They also overexpress RANKL and promote formation of large osteoclasts by fusion of pre-osteoclasts and osteoclast-committed macrophages [13]. MSCs, like osteoblasts, form interconnected networks where impaired direct cell–cell communication through connexin43 channels can contribute to driving pathological osteoclast activation, similar to that of GJA1 point mutations in oculodentodigital dysplasia (ODDD) [14,15]. Furthermore, functioning connexin43 hemichannels in osteoblasts and osteocytes can protect them from apoptosis upon bisphosphonate (alendronate) treatment, which inhibits osteoclasts activity [16].

## 2. Cytogenetics of GCTB

Telomeric associations (TAs) are frequent chromosomal aberrations, present in 50% to 70% of the cases [17,18]. A study concerning telomeric-associated proteins demonstrates the presence of a structural telomere protective-capping mechanism in the neoplastic cell [19]. Despite the presence of TAs, this capping mechanism plays a pivotal role in successful and stable telomere maintenance. The finding of capping is confirmed by a more general principle of alternative telomere lengthening promyelocytic leukaemia bodies (APBs). These capping structures are of importance for telomere maintenance since they consist of sheltering proteins as well as proteins involved in DNA damage repair (DDR) and homologous recombination [20]. As a sign of somatic genetical instability, TAs are generally found in high-grade neoplasms such as bladder cancer or osteosarcomas. Additionally, TAs are noted during key cellular processes such as mitosis and meiosis. To prevent genomic instability and damage, these aberrations should be resolved before cell division will take place [21]. Recently, a mutation in histone H3.3 (H3F3A) was identified in GCTB, which is exclusively restricted to the neoplastic cell. Immunohistochemical detection of the mutant protein is of major diagnostic help in differentiating GCTB from other giant-cell-rich bone lesions [22,23]. Other studies investigating genomic instability in GCTB show a more pronounced centrosome amplification and aneuploidy in the recurrent and metastatic cases of this disease [24,25].

## 3. Histones

Compaction of DNA into chromatin is not only efficient to fold all content into a small nucleus, but it also controls the regulation of gene expression. The basic repeat units of chromatin are nucleosomes. The nucleosome core particle includes a hetero-octamer of histones compromising of a tetramer of (H3-H4)_2_ flanked by two dimers of H2A-H2B. It is in this structure where around 147bp of DNA is finally wrapped [26]. Thus, histones are the main protein core component of chromatin and are highly conserved in eukaryotes [27]. Nucleosomes allow genomic compaction and enable important intracellular processes to operate on DNA. In short, these modifications are part of the epigenetic ‘code’ which is essential for the regulation of gene activity. The N-terminal tails of histones are modulated by a large variety of covalent posttranslational modifications such as methylation, acetylation or phosphorylation [28]. Another means of histone modulation is the embodiment of histones, H2A, H2B and H3 variants [29]. These modifications form unique readable recognition sites for proteins interpreting the epigenetic code [30]. Canonical histones are most abundant present during the S-phase in order to secure the critically needed amount of histones during replication. Histone chaperons are regarded as the helping escort proteins since they manage the delivery of histones together with their incorporation into DNA [31].

### 3.1. Histone H3.3

H3.3 is encoded by two genes, *H3F3A* and *H3F3B*. The different H3 proteins vary by their amino acid changes which are situated in the core of the histone protein. In this way, these changes regulate the binding specificity to chaperone proteins. Soluble histones associate with chaperons, and by this, chaperons control the incorporation of histones into nucleosomes [32,33,34,35]. In normal circumstances, the canonical histone H3.3 variant differs significantly from canonical H3.1 and H3.2 as it is expressed in all stages of the cell cycle. Canonical histones can be replaced by non-canonical histone variants under certain specific circumstances such as during repair of DNA damage or transcription. This replacement is essential for the maintenance of the overall chromatin integrity. It has particularly been proposed that histone H3 replacement by H3.3 results in an intrinsically less stable nucleosome and promotes transcription [36,37]. The replacement histone function of H3.3 is apparent in situations of DNA damage together with alterations causing nucleosome-depleted regions [38,39]. In this way H3.3 plays a role as a nucleosome gap-filler [40]. For example, in UV-induced DNA damaged sites H3.3 deposition is critical for replication fork progression [41].

Before deposition into chromatin, H3.3 is post-translationally modified by presumably K9 mono- and demethylation (H3.3K9me1 and H3.3K9me2), and K9 and K14 acetylation (H3.3K9ac and H3.3K14ac). Acetylation of lysines on histones results in a global lower positive charge and therefore in a weakening of the histone–DNA interactions. As a result, chromatin is more easy reachable for transcriptional activation. Histone H3 acetylation marks that are typically associated with transcriptional activity are the lysines 9, 14, 18, 27 and 56 [42]. Repressive methylation and hypoacetylation marks have also been described within telomeric heterochromatin, yet little known about the tension of these marks [43]. It is of interest that the cyclinB1–cdk1 complex catalyses mitotic cell division by activating the microtubule assembly and chromatin and DNA relaxation for increased gene transcription through phosphorylating H1 and H3 histones [44]. Besides this, incorporation of H3.3 into chromatin can effectively be done autonomously from DNA replication. H3.3 is mainly loaded at actively transcribed or euchromatic sites by the H3.3-specific chaperone HIRA [32]. Interestingly, it was demonstrated that G34 mutations hinder this type of chaperone binding [45].

### 3.2. H3.3 Histone and Telomeres

Generally H3.3 is a marker of active genes, but the death-domain-associated protein (DAXX) in complex with the transcriptional regulator α-thalassemia/mental retardation syndrome X-linked (ATRX) forms a specific chaperone that can deposit H3.3 at heterochromatic or silent regions of which pericentromeric and telomeric regions are a part. In this way H3.3 is loaded independently of the HIRA chaperone by the ATRX/DAXX chaperone [33,34,35,46,47,48,49,50,51,52]. ATRX, like DAXX, can further associate with promyelocytic leukaemia nuclear bodies (PML-NBs) and has been proposed to contribute to DAXX/H3.3/ATRX associates with pericentric heterochromatin regions that are transcriptionally silenced by H3K9 trimethylation (H3K9me3) [53]. On the other hand, DNA methylation losses of centromeres pushes the DAXX/ATRX complex directly to telomeric and subtelomeric regions in mouse ES cells. As a result, in the absence of chromatin methylation, telomeres are protected by the DAXX/ATRX complex. In this point of view, DNA methylation is of true importance in the telomeric enrolment of DAXX/ATRX. Thus, in H3.3-deficient cells the maintenance of heterochromatin, and hereby the telomeric integrity is impacted. Moreover, H3.3-deficient cells show a higher incidence of telomeric DNA damage and chromatin instability. In turn, re-expression of WT H3.3 in these cells partially restores the H3K9me3 heterochromatin mark. Recently, evidence was provided for the association between H3.3 availability and H3.3K9 trimethylation and ATRX recruitment in order to maintain the heterochromatic state of telomeres, and by that for an adequate telomere function [54]. Taken together, H3.3 functions as a ‘stand in’ at nucleosomal sites that are disrupted, such as actively described regions or damaged sites [38,39,55].

### 3.3. H3.3-G34W, Telomere Biology and GCTB

H3 histone mutations were first identified in paediatric high-grade glioblastoma in the cerebral cortex. The most important characteristics of these mutants are that they are somatic, are involved in just a single gene and occur within a specific disease at frequency [56,57]. These characteristics have led to the term “oncohistones” and acceptance of the fact that these are called ‘drivers’ of tumorigenesis [58]. Specifically, H3.3 pinpoints the promotors and also the gene bodies of genes that have actively been described [46].

In GCTB, G34 substitutions to tryptophan (G34W) were found most frequently and, to a lesser extent, substitutions to leucine (G34L) (>90%) in H3.3 encoded by *H3F3A* [59]. Moreover, in a cancer syndrome that includes pheochromocytomas, paragangliomas and GCTB, G34W mutations were demonstrated. In this syndrome it is hypothesised that the G34W mutation arises post-zygotically, rather than somatically [60]. However, in GCTB other than in this cancer syndrome, the mutation has exclusively been detected at the somatic level [19]. Moreover, recent findings indicate that H3.3-G34W is physically incorporated into chromatin, and it is suggested that the single-residue alteration of H3.3 generates epigenomic modifications with major implications for the development of the neoplastic MSCs and more generally for the neoplastic process [61].

The exact role of H3.3 mutations in the genomic stability of GCTB, and more specifically in the presence of TAs, has not been extensively studied in GCTB. As an adequate telomere function requires the conservation of a well-functioning chromatin landscape, a disruption of its structure by mutations or changed expressions of chromatin-modifying factors has been associated to dysfunctional telomeres, and hereby genomic instability [62,63,64]. One could hypothesize that a mutation in the canonical histone H3.3—resulting in a new histone H3.3 variant—seriously affects the integrity and function of chromatin [65,66]. Moreover, this phenomenon could induce major complications in the balance of gene-expression programs, and more specific in its cell fate [29]. Furthermore, changed patterns of histone acetylation and transcriptionally repressed chromatin are linked to an inadequate telomere maintenance mechanism [43].

It was shown that H3.3 G34W stromal cells exhibit a genome-wide reduction of DNA methylation (20%). In addition, segmenting the genome into large methylation domains (LMD) of more than 20kb showed that adjustments in global DNA methylation take place in LMDs associated with facultative and constitutive heterochromatin. Here H3.3 G34W joins with these heterochromatic defects contributing to the known genomic instability in GCTB [24]. This observation fits well with the structural chromosomal rearrangements other than TAs including the presence of dicentric chromosomes in GCTB [15]. Undeniably, dicentric chromosomes occur after double-strand-break formation or after telomere shortening, or due to telomere dysfunction resulting in the generation of dicentric telomeres. Subsequently, the appearance of dicentric centrosomes can directly contribute to cellular transformation and heterogeneity [67].

In GCTB, DAXX/ATRX is recruited to telomeres, indicating loss of DNA methylation which can immunohistochemically be confirmed by a low expression of the H3K9 trimethylation mark in its stromal cells. In parallel, H3.3-G34W induces hypomethylation of heterochromatic regions [58]. Therefore, cells with a decreased DNA methylation profile may be more susceptible to a DAXX/ATRX dysfunction [68].

In parallel, DAXX interacts with PML and is therefore found in PML-NBs [69,70]. PML-NBs can easily be demonstrated in GCTB stromal cells [19]. ATRX, like DAXX, associates with PML-NBs and thus contributes to the DAXX/H3.3 targeting to chromatin by binding histone-repressive marks in heterochromatin and G-rich DNA repeats [49,51,71,72]. PML links to the histone-loading machinery and in this way it modulates chromatin remodelling and cancer pathogenesis [73].

Next to H3.3′s known strong association with ATRX mutations and alternative lengthening of telomeres (ALT), it was recently shown that ATRX binds G-quadruplex (G4) DNA [24]. In general, telomeric G4-enriched sites are depleted or disrupted in nucleosome assembly [74,75,76,77]. Transcription of these repeats causes chromatin disruption and may therefore be responsible for H3.3 deposition by ATRX [78,79]. In normal circumstances, ATRX-mediated loading of H3.3 is mandatory for the rechomatinisation of the ‘late-replicated telomeric DNA’ [80,81]. In ALT cancers, the absence of functional ATRX accounts for the incompetency to rechromatinise telomeric DNA which results in DNA damage, ALT and finally, genomic instability [82]. In GCTB, however, a morphological but non-functional ALT signature was illustrated by the presence of large PML bodies accompanied by non-heterogenous lengthened telomeres. Telomerase was shown to be active together with a slight reduction in telomere lengths, which emphasises GCTB’s neoplastic characteristics next to the presence of H3.3 mutations.

In the absence of ATRX and functional H3.3, nucleosomal disruption within telomeric repeats will be present at these sites for a longer time. This consequently leads to chromatin de-repression and damage as might be the case in the telomeres of GCTB. This phenomenon was practically demonstrated in H3.3-null cells in H3.3 knocked-out mice, where the decrease in trimethylation at telomeres suggested a role for H3.3 in its maintenance. Loss of expression of ATRX cannot be demonstrated, a finding which is in line with earlier data that ALT is not activated in GCTB [11].

Telomeres are enriched in the heterochromatin histone trimethylation marks H3K9me3 and H4K20me3, as well to HP1 binding [83]. Low degrees of H3K9 trimethylation and high amounts of H3 and H4 acetylation are involved in telomeric instability. A recent study demonstrated a genome-wide hypomethylation in GCTB that affects telomeres and centromeres [58]. In view of this, nucleosomes block the binding with nucleosomal telomeric sequences and affect the interaction on adjoining linker DNA, favouring the binding of TRF-1. TRF1 binding is modulated by nucleosomes via the histone tails suggesting that changes in the chromatin structure of telomeres may influence the accessibility of TRF proteins to their binding sites [84]. A recent chromatin immunoprecipitation-exonuclease (ChIP-exo) study in yeast demonstrated that the histone H3 tail can interact with linker DNA [85]. Of interest is that these interactions are inversely influenced by H3K36me. Surprisingly, the presence of chromatin-incorporated H3.3-G34W was not accompanied by changes in the global amount of H3K36me in GCTB as was observed for other substitutions affecting H3.3 lysine 27 and lysine 36 in other neoplasms [58,86,87,88]. The H3.3 nucleosomes in GCTB accommodate H3.3-G34W as well as wild-type H3.3, and therefore mask in *cis* (on the same histone tail) effects on the H3.3-G34W terminal tail [45,58]. This combination of mutated and wild-type H3.3 in the nucleosomes of GCTB is supported by the recent finding that mutant G34W contributes only 2.6% to the total H3 pool and 25.7% to the H3.3 pool. In other words, a possible loss of the telomeric H3K36me mark could favour the binding of TRF-1 to telomeres. Additionally, overexpression of TRF-1 generally results in mainly sister–sister TAs as can be seen in GCTB. In contrast, these TAs are affecting just particular telomere ends in GCTB which may indicate local peri-telomeric differences in its telomere biology. One could suggest that a different mixture of H3.3-G34W/wild-type H3.3 in the H3.3 histones associated at the local telomere level may also result in a different H3K36me3 mark profile per telomere (very rich or poor in G34W). In parallel, recent research has demonstrated that H3-G34W substitutions block the normal working of the histone methyltransferase SETD2, which ensures the H3K36me3 marks, in *cis*, therefore, affecting the H3.3-G34W terminal tail and the approachability of TRF proteins to their individual telomere binding sites. The fact that these TAs are not clonal in the beginning of GCTB’s neoplastic process, fits well with this.

Besides their heterochromatic environment, transcribed telomeres give rise to long non-coding UUAGG-repeat transcripts also known as telomeric repeat-containing RNAs (TERRAs) [78,79]. TERRAs are transcribed from subtelomeric regions towards the telomeric ends and bind to telomeric chromatin both in *cis* and *trans* [89]. In this perspective, TERRA is a true part of the immediate environment of telomeres. TERRAs are crucial in the maintenance of telomere length and by this in telomere protection. Although an interaction between TERRAs and H3K9me3 and HP1 has been demonstrated, a more recent study with human cell lines observed that the non-coding RNA TERRA and the formation of the histone trimethylation marks H3K9me3 and H4K20me3 at telomeric heterochromatin are related [90,91]. This study also showed that the majority of the TERRAs were given birth from one single region located on chromosome 20q. This finding is of major interest in the context of this review as a study already validated a 20q11.1 amplification together with overexpression of TPX2 as a candidate oncogene in GCTB [92]. However, depletion of TERRAs causes a firm reduction of the H3K9me3 and H4K20me3 marks at telomeric chromatin. Therefore, TERRA is of importance in the management of histone methylation, and moreover in the telomeric heterochromatin assembly [78]. Interestingly, when protein HP1α is targeted to a telomere, a reduction of telomerase activity is induced. Meanwhile, an increased heterochromatinization takes place which finally leads to a higher overall telomere protection [93].

Genes adjacent to the telomeric regions often show a low expression and are part of transcriptional silencing, which is also called the telomere position effect (TPE) [94]. Recent studies reported that genes at a much greater distance from the subtelomeric regions can be affected by the TPE which is called TPE over long distance (TPE-OLD). This phenomenon is due to the three-dimensional location of telomeres relative to otherwise transcriptionally active genes [95,96]. In the presence of telomeric t-loops, genes located at greater distance from the telomere ends are affected by silencing [97]. This phenomenon affects several genes, including the hTERT gene in humans [96,98,99]. As a consequence of telomere shortening, this silencing loop structure is no longer adjacent to hTERT, which results in hTERT transcription and telomerase activity. In GCTB, telomerase activity together with shortening of telomere ends is reported, which may indicate that TPE-OLD is affected. As a result, some extra (morphological) genes may be re-activated in the mononuclear neoplastic cell.

### 3.4. H3.3-G34W and Malignant GCTB

Sarcomatous transformation of a conventional GCTB (secondary malignant GCTB) is a rather uncommon phenomenon accounting for an incidence of 2.4% of all GCTB cases [100]. Most cases occur after radiotherapy, although occasionally malignancy may arise subsequent to denosumab treatment [100,101,102,103]. The malignant part of the otherwise conventional type GCTB does not have specific morphological features. This transformation may be into an undifferentiated sarcoma or into several types of osteosarcoma [100,104]. (1,5) Primary malignant GCTB is even more rare and accounts for 1.6% of all GCTB cases [100]. In some of these cases the H3.3-G34W mutation is retained [100,104]. When dealing with a giant-cell-rich osteosarcoma the differential diagnosis with primary or secondary maligant GCTB can be very challenging. Next to histological examination, interpretation of the radiological features may be of major help in distinguishing both entities. Molecularly, a recent publication reported that maligant bone tumours with a H3.3-G34W mutation possess features resembling osteosarcoma [105]. More specifically, they carry an increased burden of somatic mutation variants. Of major interest in view of the differential diagnosis with (giant cell rich) osteosarcomas, and more specifically concerning the relation between H3.3-G34W and telomere biology in GCTB, is that H3.3-G34W mutated maligant bone tumours are enriched with a variety of alterations involving *TERT* suggesting telomere dysfunction in the transformation of benign to malignant GCTB [105]. This finding is in line with the above-stated hypothesis of a delicate balanced telomere stability in GCTB. Once instable, futher cytogenetical abberations are needed in order to transform into malignancy where the H3.3-G34W mutation itself may become nonessential in this process. Moreover, the authors found that hypermethylation of the *CCND1* promotor is specific for GCTB and therefore cyclin D1 is a plausible cancer-driver gene. This finding is in line with the earlier-mentioned accelerated cell cycle in GCTB where its progression is linked to increased risk of disease progression [41].

### 3.5. H3.3-G34W, Histomorphology and Denosumab

The GCTB stromal cell displays markers of both the mesenchymal stem cell as well of the pre-osteoblast [19]. Tumour stem cells (TSC) are clearly present in a diversity of tumours, including GCTB. Next to their ability to form spheres, to be multipotent, to display a rapid growth and to develop multidrug resistance, these cells express the Stro-1 antigen [106]. In addition, there is evidence that the neoplastic MSCs are of osteoblastic lineage in GCTB. This is illustrated by a downregulated expression of genes such as stathmin-like 2 (STMN2), IGF2 and LEP in H3.3 G34W stromal cells. These genes are known to be involved in mesenchymal stem cell differentiation and murine osteoclastogenesis, and therefore in the endorsement of bone differentiation [58,107,108]. Additionally, the stromal cells of GCTB express (pre)osteoblast markers, such as RUNX2, SATB2 and OSTERIX [109]. So epigenetic regulation is closely related to bone homeostasis, whereas histone modifications are implied in the control of cell differentiation [110,111,112]. In general, when cells are differentiating and maturing, they may lose pluripotency characterised by epigenetic changes in order to shut off genes dealing with stemness [113]. In this process H3K4me3, H3K27me2/3, H3K79me2/3 and H3K9me2/3 residues are involved in order to control gene expression and to reprogram the cell’s fate [114]. In this point of view, stem cells do not show relevant histone modifications of H3K79me2/3 and H3K9me2/3 [115,116]. These phenomena of epigenetic modifications, such as H3K9me3, H3KM27me3 and H3K4me3, direct optimal osteoblastic differentiation of mesenchymal stem cells. Additionally, H3K36 trimethylation is increased during osteogenic differentiation [117]. In other words, high levels of H3K27me3 and H3K9me3 can be found in genes responsible for embryonic stem cell maintenance, while H3KM36me3 must increase during osteogenic differentiation. Interestingly, trimethylation of H3K36 is mediated by SETD2, thus the fate of mesenchymal stem cells is finally controlled by SETD2 [118]. As recent research demonstrated that G34W substitutions in H3.3 block SETD2′s activity, this mechanism could explain the cell fate of the mononuclear spindle-shaped cells in GCTB.

Although G34W mutations occur on a residue which is not involved in posttranslational modifications, it seems that the adjacent K36 residue is spherically affected, leading to a decreased H3.3K36 trimethylation [87]. Moreover, this leads to a mutual increase of H3.3K27 trimethylation on the mutant tail [46]. The earliest event following the H3.3K36me3 loss on G34W histones is the deposition of the H3K27me3 mark on the mutant tail in active gene regions that are now enriched for H3.3 as well for G34W. The loss of the H3.3K36me3 mark in geneic regions results in H3K27me3 redistribution from lower-affinity intergenic regions to promotor and geneic regions probable by the PCR2 complex. This redistribution initiates transcriptional adjustments in a background without epigenetic remodelling. The G34W-mediated epigenetic remodelling affects the mesenchymal cell lineage commitment. As an example, genes involved in muscle functions are downregulated by G34W, thereby suggesting that farther differentiation into a mesenchymal myogenic progenitor cell is blocked [119]. The presence of smooth muscle actins in GCTB’s stromal cells had already been proven [120].

When treated with denosumab, which is a human monoclonal antibody directed against the receptor activator of nuclear factor-κB (RANKL), the mutant cells mature and form woven bone [121,122,123,124]. This process reverses on withdrawal of treatment. Depletion of osteoclast-like cells implies that an osteoclast-like growth factor suppresses stromal cell differentiation and results in proliferation. Apparently, the use of denosumab initiates a differentiation process into a more osteoblastic-committed cell population in GCTB. It is well known that pre-osteoblasts consistently express GATA4, and that this is gradually downregulated during osteoblast differentiation [125]. To the best of our knowledge, GATA4 expression in GCTB and denosumab-treated GCTB has not been studied so far.

An important feature of GCTB is that H3F3A mutant cells survive denosumab treatment [126]. Among the gene expression profiles in H3.3G34W mutated stromal cells E2F targets are the most downregulated genes. This finding suggests that the high proliferative activity of these mutated stromal cells may be a consequence of silencing negative regulators of the cell cycle, most probably the E2F4-6 family members. When taking the relevant genes of GCTB which are targeted by E2F transcription factors into account, RANKL and osteoprotegrin (OPG) were found to be differentially targeted. Concerning OPG, the transcription start site is addressed by E2F while the RANKL site lacks association with E2F. This mechanism indicates a cooperation between the H3.3 and E2F transcription factors. Furthermore, the G34W mutation leads to a loss of E2F control resulting in secretion of high levels of RANKL together with reduced levels of its decoy receptor OPG [127]. RANKL can be found as a membrane-anchored but also as a soluble molecule both acting as agnostic ligands for RANK. Creating an uncontrolled osteoclastogenesis, as is the case in GCTB, soluble and diffusely available RANKL is needed in the tumour microenvironment. Following proteolytic cleavage by matrix metalloproteinases such as matrix metalloproteinase 14, RANKL is in this way released from the cell surface [128]. One may expect that MMP14 is overexpressed in GCTB, a finding that was confirmed by a study designed to investigate the role of extracellular matrix turnover in GCTB. In contrast, targeting RANKL with denosumab could have a negative effect on the tumour vasculature as RANKL is known to regulate endothelial cell survival through expression by arterial smooth muscle cells [129]. As a result, fibrosis and a loss of vasculature is observed in GCTB when denosumab is applied.

The fact that GCTB is kept under control using denosumab raises new questions about the effect of this treatment, specifically on the cell biological behaviour of the neoplastic stromal cells. A possible explanation and a better understanding of this may be found in the signal mechanisms of coupling of bone resorption and formation, a process that is logically uncoupled in the tumour microenvironment of GCTB. These signals are transmitted by osteoclasts to osteoblasts in order to drive the transition from bone resorption to formation [130]. A recent study demonstrated that the RANKL-binding peptide drugs WP9QY and OP3-4 not only inhibit osteoclastogenesis, but surprisingly also stimulate osteogenic activity in osteoblasts [131]. The candidate ligand would be osteoblastic membrane-bound RANKL, which was recently demonstrated in osteoblasts from mice [132]. Therefore, the partners that bind osteoblastic RANKL would be RANK or the negative regulator of RANKL-forward-signalling osteopontegerin (OPG) [133]. Research performed on osteoblasts in OPG-deficient mice showed no reduction of osteoblastic osteogenic activity when using osteogenic growth peptide. Therefore, OPG is not regarded as a ligand in the RANKL reverse signalling [134]. RANK was tested to determine whether it would activate this signalling. An important conditional finding was that RANK must be crosslinked to activate an effective signalling in the osteoblast. Moreover, it was shown that multimeric assembly of RANKL by vesicles is necessary to trigger the signalling illustrated by an upregulation of Runx2 and osterix. Thus, mature, and active osteoclasts secrete vesicular RANK to activate this signal pathway [119]. This is an important finding because this is an essential part in understanding the effect of denosumab on GCTB cell biological behaviour. As denosumab is not crosslinked, this antibody can bind RANKL and inhibit osteoclastogenesis, but it is not potent enough to activate the RANKL-reverse signalling pathway in the neoplastic mononuclear cells. Hereby, the only option left is secretion of vesicular RANKL by osteoclasts. Treatment with denosumab has an indirect effect on the neoplastic cells as secretion of vesicular RANKL by osteoclasts is stopped by inhibition of osteoclastogenesis mainly by neutralising RANKL that had been secreted by the neoplastic mononuclear cells. On the other hand, suppression of RANKL reverse signalling, by, for example, a reduction of vesicular RANKL due to denosumab treatment, may lead to an inefficient bone formation [119]. This is in line with previous reports studying the same effect of Runx2 expression in osteoblasts [135].

## 4. Summary

The recent finding of a highly recurrent mutation in histone H3.3 has opened new molecular insights concerning the etiogenesis of GCTB. While this driver mutation is highly conservative and unique, much attention has been paid to its histone biology, and more specifically in relation to H3.3 mutations in glioblastoma *et vice versa.* Here, we focus on the possible relations between the G34W mutant stromal cell and other cell biological features which are characteristic for GCTB, such as the presence of TAs, its slight cytogenetical instability, the reactivation of hTERT, and its loss of control of RANKL secretion (see Figure 3). 

As part of the H3.3 mutations, the key features of the histone mutants in general are that these mutations are somatic, that they occur just in a single gene, and that they arise at high frequency within a particular disease. In addition, there is a close affective link between a disturbed histone biology and the transcriptional activity of chromatin. In view of a delicate balanced protective telomere biology in GCTB, characterised by PML-NB/Daxx/ATRX telomere capping to prevent TAs, H3.3-G34W induces a global hypomethylation of chromatin. This could lead to an affected telomere integrity and instability where the nucleosome ‘gap-filling’ is not ensured anymore which results in telomere shortening. This is the trigger to allocate ATRX/Daxx to the telomere ends associating with PML-NBs as part of the histone loading machinery in these structures to ensure proper methylation of the affected chromatin. In addition, hypomethylation of telomeres due to G34W decreases the amount of TERRAs and reactivates hTERT to compensate for the telomere length loss. As H3.3 histones in GCTB are variably loaded with H3.3-G34W and wild-type H3.3, a variable affection of the H3.3-G34W terminal tails is present on each telomere. The accessibility of TRF proteins to their binding sites could hereby be affected resulting in non-clonal TAs, especially at the beginning of GCTB’s neoplastic process. Another characteristic feature of GCTB is the presence of very large and evenly distributed osteoclastic giant cells strongly indicating that RANKL is overexpressed in the tumoral stroma while the negative feedback loop of OPG is blocked. G34W decreases the gene expression of E2F targets. As a result, it was objectivated that this results in a differential loss of control of an enhanced RANKL and decreased OPG gene expression in GCTB. Moreover, MMP14 overexpression has been demonstrated in GCTB responsible for cleavage of membrane-bound RANKL into soluble RANKL to enhance the uncontrolled osteoclastogenesis locally.

While denosumab is effective in breaking the vicious circle of osteoclastogenesis, G34W mutant stromal cells will not be affected in their survival by its administration. The conserved altered H3.3 nucleosome biology together with its effects on chromatin methylation, cytogenetic and telomeric instability, and uncontrolled differential RANKL/OPG transcriptional activity still remains the base for recurrent disease.

## 5. Conclusions

GCTB is characterized by a complex and difficult-to-understand cell biological behaviour because of its heterogenous morphology, the different downward effects of the nucleosome H3.3 mutation, its uncontrolled osteoclastogenesis and its delicate balanced telomere stability which could pave the way to malignancy. While denosumab is effective in breaking the vicious circle of osteoclastogenesis, G34W mutant stromal cells will not be affected in their survival by its administration. The conserved altered H3.3 nucleosome biology, together with its effects on chromatin methylation, cytogenetic and telomeric instability, and uncontrolled differential RANKL/OPG transcriptional activity still remains the base for recurrent disease.

## Figures and Tables

**Figure 1 cancers-13-05119-f001:**
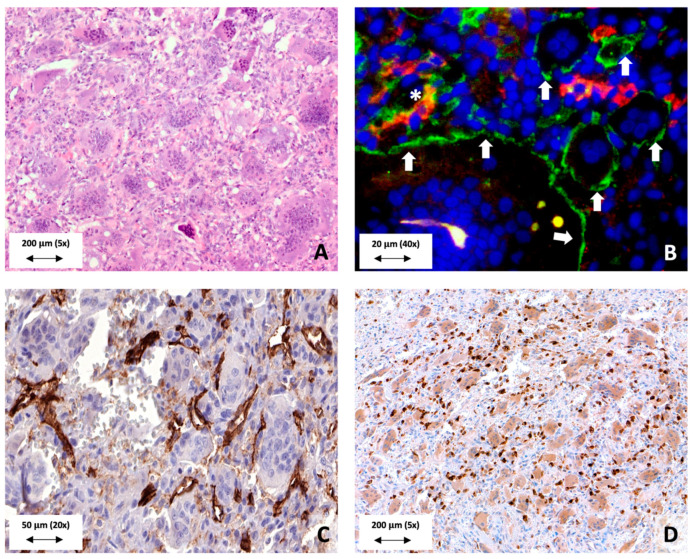
Morphological aspects of GCTB. (**A**) GCTB is composed of four major components. These are large osteoclastic giant cells, macrophages, mononuclear spindle-shaped cells and the extracellular matrix. (**B**) Immunofluorescent membrane expression of CD33 (green) by osteoclasts (white arrows). Some macrophages express CD14 (red) on their membrane (asterix *). Note that other macrophages express CD14 as well CD33 on their membrane. (**C**) CD31 immunohistochemistry demonstrates the presence of many intratumoural blood vessels. Note that almost every large osteoclastic giant cell is connected to at least one blood vessel branch. (**D**) RANKL expression by the neoplastic mononuclear spindle-shaped cells in order to enhance osteoclastogenesis.

**Figure 2 cancers-13-05119-f002:**
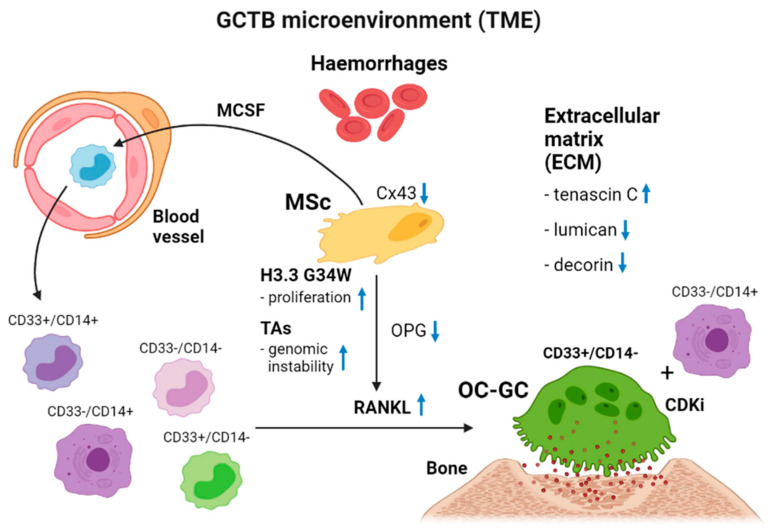
Schematic representation of the tumour microenvironment of GCTB. MCSF, macrophage colony stimulating factor; MSc, mononuclear spindle cell; TAs, telomeric associations; TME, tumour microenvironment; ECM, extracellular matrix; OPG, osteoprotegrin; OC-CG, osteoclastic giant cell. The interference of the four different components of GCTB makes the biological behaviour of GCTB unpredictable. The mononuclear spindle-shaped cell as first component harbours an H3.3 driver mutation affecting several internal and external cell biological processes. By secretion of MSCF, blood-borne monocytes are attracted to the tumour microenvironment. Under the influence of extensive RANKL in combination with a lower OPG secretion by the neoplastic cell, high numbers of large osteoclastic giant cells (second component) are formed by fusion of CD33^+^/CD14^−^, CD33^+^/CD14^+^ and CD33^−^/CD14^+^ macrophages. These osteoclastic giant cells are responsible for the osteolytic properties of GCTB. Macrophages not committed to osteoclastogenesis are regarded as tumour-associated macrophages (third component). The extracellular matrix (fourth component) facilitates the mobility of cells in the matrix by expressing tenascin and underexpressing decorin and lumican. The presence of haemorrhages is mostly seen in primary tumours that behave more aggressively or that give rise to benign metastatic lung plugs. Finally, a circulus vitiosus of osteolytic expansion has been created by the tumour itself.

**Figure 3 cancers-13-05119-f003:**
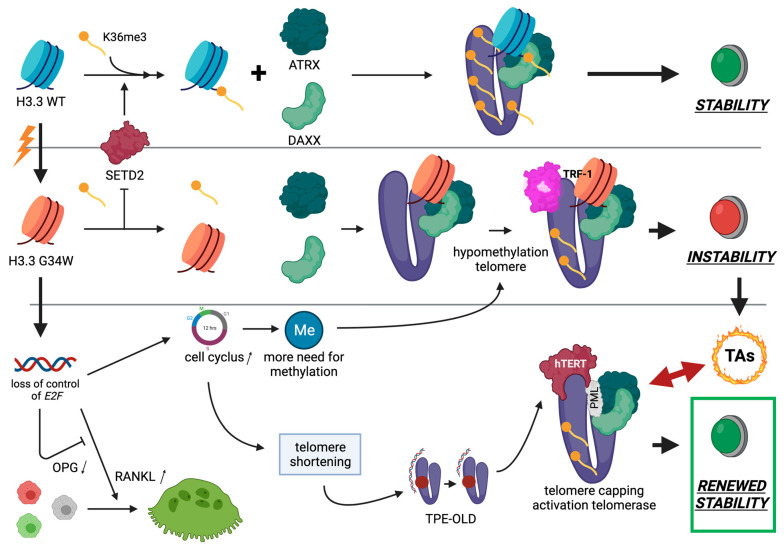
Cell biology of GCTB. See left upper corner. Mutation leading from wild-type H3.3 (blue) to H3.3G34W (orange) inhibits the activity of the histone methyltransferase SETD2, which is in normal circumstances responsible for H3K36me3 of the histone tail. Wild-type H3.3 telomeres (purple) are sufficient methylated, and form together with ATRX and DAXX stable telomere ends. In contrast, affecting the H3.3-G34W terminal tail leads to a relative hypomethylation status. The loss of the telomeric H3K36me mark could favour the binding of TRF-1 to telomeres resulting in a relative telomeric instability further leading to mainly sister–sister TAs. In H3.3G34W mutated stromal cells E2F targets are the most downregulated genes. This suggests that the high proliferative activity of these mutated cells may be the result of silencing negative regulators of the cell cycle, most probably the E2F4-6 family members. When taking the GCTB-relevant genes targeted by the E2F transcription factors into account, RANKL and osteoprotegrin (OPG) are differentially targeted which leads to a loss of E2F control resulting in secretion of high levels of RANKL together with reduced levels of its decoy receptor OPG, and therefore to an enhanced osteoclastogenesis. Due to a higher proliferative activity there is more need for methylation of the telomeres, while these hypomethylated telomeres shorten faster. As a result of this shortening, the silencing loop structure of TPE-OLD is no longer in the proximity of hTERT, leading to hTERT transcription and telomerase activity. Together with the formation of a structural telomere protective cap (PML/ATRX/DAXX) this whole mechanism plays a pivotal role in successful and stable telomere maintenance leading to a renewed telomeric and hereby genomic stability.

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
