# Peer review of "Cell Biology of Giant Cell Tumour of Bone: Crosstalk between m/wt Nucleosome H3.3, Telomeres and Osteoclastogenesis"

_cancers, 2021, doi:10.3390/cancers13205119_

Round 1

Reviewer 1 Report

Dr. Forsyth et al present in this manuscript entitled “Cell biology of giant cell tumor of bone: crosstalk between mutated and wild-type nucleosome H3.3, telomere maintenance and uncontrolled osteoclastogenesis”, a biologic route trying to harmonize the main driver of GCTB, the characteristic mutation involving G34 in the neoplastic spindle cells, with other recurrent biologic findings in this tumor as telomerase association, some degree of cytogenic instability, and reactivation of hTERT.

The authors infer the correlations supported by the literature and launch plausible well-reasoned hypotheses. The manuscript is well written, and it is more a compilation, it is an integration of key biological dysfunctions in GCTB. It deserves to be published.

I have some comments that should be considered as minor:

1.- Authors presented different key components of GCTB (osteoclast related and no related macrophages, extracellular matrix components, and spindle mononuclear malignant osteoblastic component) but in the manuscript, very few data have been presented trying to integrate these different components in the GCTB genesis or maintenance. It maybe should be explained that the focus of this review will be the genomic and epigenomic alterations of the malignant component.

2.-  Could the authors make suggestions about the biologic underlying processes in the malignant transformation of GCTB?

3.- Wording correction: “Los” in row 264; “IN” in row 248.

Author Response

Dear reviewer, 

Thank you for your much rewarded review report. Here are the answers / adaptions:

  1. We made a comment in lines 18 till 21 stating that the genomic and epigenomic alterations that may lead to genomic instability and malignancy are in focus of our review.
  2. We added a small paragraph in the manuscript entitled 'H3.3-G34W and malignant GCTB' dealing with biological underlying processes in the malignant transformation of GCTB (rules 269-288).
  3. We made the corrections accordingly

Reviewer 2 Report

The article entitled “Cell biology of giant cell tumour of bone: crosstalk between mutated and wild-type nucleosome H3.3, telomere maintenance and uncontrolled osteoclastogenesis” the authors Forsyth et al., reviewed on GCTB microenvironment, cytogenetics, role of H3.3 histone, its mutated form H3.3-G34W and telomere biology. Further authors explained possible relations between the G34W mutant stromal cell and telomeric associations, cytogenetical instability and reactivation of hTERT, and loss of RANKL secretion. Finally, authors discussed on available targeted therapy and the basis of GCTB recurrence.  

Overall, it is a timely review and interesting to the readers of this journal. However, below comment to be addressed before considering for publication.

Comments:

  1. Consider a shorter title.

Author Response

Dear reviewer, 

Thank you for your much rewarded review report.  

We shortened the title into: "Cell biology of giant cell tumour of bone: crosstalk between m/wt nucleosome H3.3, telomeres and osteoclastogenesis".